# Long-Term Neurological and Psychological Distress Symptoms among Smallholder Farmers in Costa Rica with a History of Acute Pesticide Poisoning

**DOI:** 10.3390/ijerph18179021

**Published:** 2021-08-26

**Authors:** Andrea Farnham, Samuel Fuhrimann, Philipp Staudacher, Marcela Quirós-Lépiz, Carly Hyland, Mirko S. Winkler, Ana M. Mora

**Affiliations:** 1Swiss Tropical and Public Health Institute, Socinstrasse 57, 4051 Basel, Switzerland; samuel.fuhrimann@swisstph.ch (S.F.); mirko.winkler@swisstph.ch (M.S.W.); 2Swiss TPH, University of Basel, Petersplatz 1, 4001 Basel, Switzerland; 3Institute for Risk Assessment Sciences (IRAS), Utrecht University, P.O. Box 80177, 3508 TD Utrecht, The Netherlands; 4Swiss Federal Institute of Aquatic Science and Technology (Eawag), Ueberlandstrasse 133, 8600 Dübendorf, Switzerland; Philipp.Staudacher@eawag.ch; 5Institute of Biogeochemistry and Pollutant Dynamics, Department of Environmental Systems Science, ETH Zürich, Universitätstrasse 16, 8092 Zürich, Switzerland; 6Central American Institute for Studies on Toxic Substances, Universidad Nacional, Heredia 83-3000, Costa Rica; mquiroslepiz@gmail.com (M.Q.-L.); chyland@berkeley.edu (C.H.); animora@berkeley.edu (A.M.M.); 7Center for Environmental Research and Children’s Health (CERCH), School of Public Health, University of California at Berkeley, 1995 University Avenue, Suite 265, Berkeley, CA 94720-7392, USA

**Keywords:** Costa Rica, pesticides, agriculture, farmers, pesticide poisoning, environmental health

## Abstract

Studies suggest that acute pesticide poisonings (APP) may be linked with long-term neurological effects. To examine long-term neurological and psychological distress symptoms associated with having experienced an APP, we conducted a cross-sectional study of 300 conventional and organic smallholder farmers from Zarcero County, Costa Rica, May–August 2016. We collected self-reported data on sociodemographic characteristics, occupational history, pesticide exposure, APPs, neurological and psychological distress symptoms (using the Brief Symptom Inventory (BSI)). Adjusted logistic regression models were fit. A total of 14% of the farmers (98% male) reported experiencing at least one APP during their lifetime. Self-reported APP was associated with neurological symptoms during the 12 months prior to interview (e.g., fainting (Odds Ratio: 7.48, 95% Confidence Interval: 1.83, 30.74), shaking hands (3.50; 1.60, 7.60), numbness/tingling in hands or feet (3.23; 1.66, 6.32), insomnia (2.53; 1.34, 4.79), accelerated heartrate (2.42; 1.03, 5.47), dizziness (2.38; 1.19, 4.72), increased irritability/anger (2.37; 1.23, 4.55), low energy (2.33; 1.23, 4.46), and difficulty concentrating (2.01; 1.05, 3.85)). Farmers who reported an APP in the ten years prior to interview experienced increased odds of abnormal BSI scores for hostility (4.51; 1.16, 17.70) and paranoid ideation (3.76; 0.99, 18.18). Having experienced an APP may be associated with long-term neurological and psychological distress symptoms.

## 1. Introduction

Occupational exposure to pesticides is ubiquitous among farmers worldwide [1] and its adverse effects on human health are a well-documented threat [2,3]. The burden of pesticides on health is particularly high in low- and middle-income countries (LMICs), due in part to pesticide applicators with less education, limited access to personal protective equipment [4,5], larger agricultural populations, and fewer or less enforced regulations around pesticide use [2,6,7]. It is estimated that 99% of deaths from acute pesticide poisonings (APPs) occur in LMICs, despite the fact that these countries only use 20% of the pesticides produced internationally [8]. The true burden of APPs in LMICs is unknown, but previous studies have indicated that as few as 5% of cases are recorded by the official national registries [7,9]. While European countries have developed legislation to restrict the use of hazardous pesticides such as glyphosate, the use of pesticides in developing countries has grown rapidly without similar regulation [1].

Quantification of the burden of APPs in LMICs is made more challenging by the fact that their clinical presentation varies by pesticide class and sometimes by active ingredient [2,7]. Poisoning by organophosphate pesticides and carbamates (acetylcholinesterase inhibiting insecticides) is associated with multiple symptoms, including headache, dizziness, bradycardia, vomiting, and paralysis, among others [2]. Poisoning by herbicides such as paraquat, on the other hand, is associated with mucous membrane and airway irritation, abdominal pain, diarrhea, and vomiting, among others [2]. The non-specificity of these symptoms means that it is difficult to conclusively diagnose an APP without immediate confirmation via a doctor visit, which is often not available in LMICs; however, previous studies have suggested that recall accuracy of exposures to pesticide classes or groups is good (range = 0.6–0.9) and that self-report of previous APPs is better than using medical records due to under-reporting [10,11,12,13,14].

Studies in both LMIC and high-income countries have suggested that there may be long-term health effects of APPs [15,16,17,18,19,20,21,22,23,24,25]. Two studies in Ethiopia and South Africa have reported associations of APPs with adverse long-term neurobehavioral outcomes, such as dizziness, sleepiness, headache, and overall neurological symptom scores [15,16]. Several studies in the United States and LMICs have also linked APPs with short and long-term psychological distress symptoms, including suicide, aggression, and depression [18,19,20,21,22,23,26]. However, disentangling chronic effects of past APPs from the ongoing effects of low-level pesticide exposure remains challenging, especially in LMICs where the burden of APPs is the highest [8,27,28]. By comparing farmers who switched from conventional farming practices (i.e., intensive use of synthetic pesticides) to organic farming practices (i.e., no use of synthetic pesticides), it is possible to assess whether lingering symptoms of past APPs remain even in those who no longer use synthetic pesticides. In our study of smallholder farmers from Costa Rica, we aimed to assess the associations of past self-reported APPs with (i) sociodemographic characteristics of farmers, (ii) frequency of neurological symptoms during the 12 months prior to the interview, and (iii) psychological distress symptoms during the month prior to interview. To our knowledge, we present the first study to look at a wide range of neurological and psychological distress symptoms associated with having experienced an APP in both conventional and organic farmers.

## 2. Materials and Methods

### 2.1. Study Design and Participants

We conducted a cross-sectional study of 300 smallholder farmers (273 men, 27 women aged 18–75) in Zarcero County, Costa Rica, between May and August 2016 [29]. This study is part of the larger Pesticide Use in Tropical Settings (PESTROP) project, which aims to assess the interactions between pesticide use-related environmental exposure, human health effects, and institutional determinants in two tropical agricultural settings (i.e., Zarcero Country in Costa Rica and Wakiso District in Uganda) [30]. The PESTROP project’s subject recruitment and procedures have been described in detail elsewhere [5,29,30]. Briefly, conventional farms in the study area were selected using a random spatial sampling methodology and smallholder land-use data. Organic farms were chosen based on a list of farms from the local organic farmers’ association or from in-person identification. Farmers were eligible to participate if they were farm owners, permanent workers, or temporary pesticide applicators aged ≥18 years who owned or worked in farms located in the study area. They were ineligible if they had a diagnosis of psychiatric disease or used psychopharmacological medications. The active ingredients most commonly applied by the farmers in the 12 months before the study visit were the fungicide chlorothalonil, the herbicides paraquat and glyphosate, and the pyrethroid insecticide cypermethrin [5]. There was high variability in weekly pesticide exposure scores both within and between workers at the time of the interview, with limited protection of hands (gloves), eyes (glasses) and airways (masks or respirators) [4].

The present study was approved by the human subjects committee of the Universidad Nacional in Costa Rica (UNA-CECUNA-ACUE-04-2016) and Ethical Board of the Ethikkommission Nordwest- und Zentralschweiz in Switzerland (EKNZ-UBE 2016-00771). Written informed consent was obtained from all study participants. The study was carried out in accordance with The Code of Ethics of the World Medical Association (Declaration of Helsinki) for experiments involving humans and the Uniform Requirements for manuscripts submitted to Biomedical journals.

### 2.2. Occupational Pesticide Exposure and Sociodemographic Characteristics

We administered a structured questionnaire to farmers to collect data on sociodemographic characteristics and occupational history. To assess previous occupational pesticide exposure, we asked farmers the age at which they started working in agriculture and using pesticides, their work history in organic or conventional farms, and if they handled/ sprayed pesticides (specifically the 15 pesticide active ingredients most commonly used in the study area in 2014–2015) [4] during the 12 months prior to interview. To assess recent pesticide exposure, we asked if they had handled or applied pesticides during the week prior to interview and the specific pesticide active ingredients. The details can be found in Table 1.

### 2.3. APPs and Associated Neurological and Psychological Distress Symptoms

We asked farmers if they had ever had an APP (“In your entire life, have you ever suffered a pesticide poisoning?”) and if they had, how many times. For each APP, we also asked when it occurred and if it was confirmed by a doctor.

All farmers (regardless of previous APP) were asked if they had experienced any symptoms of APP (e.g., excessive salivation, lacrimation, vomiting, diarrhea) during the 12 months prior to interview (see Figure 1 for complete list of symptoms). The list of 31 symptoms was based on previous studies of Latin American farmworkers [22,31,32]. The farmers were asked to report the frequency of the symptom in the last 12 months on a scale from 0 to 4 (never, once in the last 12 months, once a month, once a week, more than once a week). Mean frequency of symptoms among different subgroups (e.g., those who had experienced an APP vs. those who had not) was then calculated by summing the total frequencies for each symptom divided by the total number of non-missing responses in that subgroup. Symptom occurrence was not used to define whether they had an APP, as clinical presentation can vary widely based on the poisoning agent and the non-specificity of symptoms makes it difficult to definitively diagnose all cases of APP [2].

We administered the Spanish version of Brief Symptom Inventory (BSI) [33] to all farmers to assess the prevalence of psychological distress symptoms in the month prior to interview. The BSI is a standardized questionnaire of 53 items that has been successfully used in Latin American populations [22,34,35]. This instrument was originally designed to be self-administered, but because many farmers had low literacy, two psychometricians administered it verbally to all study participants using a visual scoring scale of 0 to 4 (never, rarely, sometimes, often, almost all the time). The BSI measures nine dimensions of psychological distress: somatization, obsession-compulsion, interpersonal sensitivity, depression, anxiety, hostility, phobic anxiety, paranoid ideation, and psychoticism. We calculated symptom dimension-specific scores dividing the subscores by the number of questions within the dimension. We also calculated the Global Severity Index (GSI), a measure of general psychological distress, by dividing the sum of scores of all BSI questions by the number of questions. We then normalized domain-specific and GSI scores to T-scores (mean = 50, standard deviation (SD) = 10), using gender-specific data from the BSI manual [33]. Lastly, we dichotomized all scores using a cut-off T-score of ≥63, which is considered a clinically relevant level of distress [33]. Psychometricians were blinded to the exposure status of the participants.

### 2.4. Statistical Analyses

We calculated descriptive statistics for all variables to describe the characteristics of the study population disaggregated by whether they had experienced an APP (Table 1).

To determine whether having experienced an APP was associated with key sociodemographic and occupational characteristics selected a priori using a causal diagram (age, sex, nationality, marital status, years of education, being a farm owner vs. a farm worker, type of farm at time of interview, having received training in safe practices for pesticide use, years of pesticide application), we first ran unadjusted logistic regression models. We then fitted a multivariate logistic regression model with explanatory variables whose p-value in the bivariate analyses was <0.20 (thus omitting type of farm at time of interview and years of education from the full model). Years of pesticide application was omitted from the model because it was highly correlated with age (Kendall’s tau = 0.78, *p* < 0.001); age was chosen to remain because it was not subject to recall bias and there was virtually no missing data. Odds ratios (ORs) and associated 95% confidence intervals (CI) were reported.

To determine whether overall mean frequency of symptoms reported differed between those who had previously experienced an APP and those who had not, we conducted a two-way analysis of variance (ANOVA) analysis. If the ANOVA results were significant, we ran a Tukey’s test to compare individual symptom frequencies while adjusting for multiple testing. We then fitted ordinal logistic regression models for the frequency of each health symptom during the 12 months prior to interview (outcome) regressed on ever having experienced an APP (yes/no, predictor) while adjusting for age, restricting to men (due to the low number of women having experienced an APP (*n* = 1)), and using the *polr* command from the MASS package in R Statistical Software. We adjusted for age and sex as they were likely confounders (associated with experiencing an APP and the frequency of symptoms). As a sensitivity analysis, the same two-way ANOVA analysis was conducted on those who applied synthetic pesticides at the time of the interview and non-applicators, to see whether the overall mean frequency of symptoms reported reflected “current” exposure to synthetic pesticides.

To model whether having experienced an APP predicted clinically significant psychological distress based on BSI scores, we fitted logistic regression models for each symptom dimension and the GSI using dichotomized T-scores representing clinically relevant distress (1: T-score ≥ 63; 0: T-score < 63). We adjusted all models for age, education, and synthetic pesticide use at the time of the interview, as these are previously reported confounders in the literature [22]. All analyses were conducted using R Statistical Software version 3.6.1 [36].

## 3. Results

### 3.1. Descriptive Statistics of the Study Population

About 14% (*n* = 43) of farmers reported at least one APP during their lifetime, with 3% (*n* = 8) reporting more than one (maximum = 10 APPs). Of the 72 APPs reported, 36% (*n* = 26) were confirmed by a doctor, 76% (*n* = 55) occurred to farmers working in conventional farms at the time of interview, and 82% (*n* = 59) were able to be approximated by the farmer to a year (time range = 1972–2015). Of the 59 APPs reported by farmers who remembered the date the event occurred, only 11 (19%) took place in the 10 years prior to the interview. All farmers working in organic farms at the time of the interview who had experienced an APP (*n* = 7/43) were working in conventional farms at the time of their poisoning.

Farmers who had experienced at least one APP were mostly male (98%), older (median age 50.0 vs. 31.0 years), had less education (completed a high school education 7% vs. 15%), had spent more years applying pesticides (median years 34.0 vs. 14.0), and were more likely to report receiving training on safe practices for pesticide use (66% vs. 45%) compared to those who had never experienced an event (Table 1). More Costa Rican born farmers reported having experienced at least one APP (18%, *n* = 32/177) than those Nicaraguan born (9%, *n* = 11/123).

Pesticide use patterns at the time of the interview were similar between those who had experienced an APP and those who had not. Both groups reported using pesticides for a median of 8 h per week and using pesticides classified by the World Health Organization (WHO) as extremely or highly toxic [37] during the week (10% vs. 15%) and the 12 months (58% vs. 65%) prior to interview.

### 3.2. Sociodemographic Predictors of Having Experienced an APP

In unadjusted models, each increasing year of age (OR: 1.05, 95% CI: 1.03, 1.08), male sex (OR: 4.73, 95% CI: 0.96, 85.51), being married or living in a partnership (OR: 2.35, 95% CI: 1.15, 5.23), having received training in pesticide use (OR: 2.62, 95% CI: 1.32, 5.21), being of Costa Rican nationality (OR: 2.25, 95% CI: 1.12, 4.85), and being a farm owner (OR: 3.36, 95% CI: 1.73, 6.68) were associated with increased odds of having experienced an APP. However, in the full model, only age remained associated with increased odds of experiencing an APP (OR: 1.04, 95% CI: 1.01, 1.08).

### 3.3. Self-Reported Neurological Symptoms During the 12 Months Prior to Interview

The most common symptoms reported were feeling tired (69%), having a headache (65%), and feeling tense (64%). Most neurological symptoms—except for loss of appetite, paleness, stomach pain, feeling fearful, and nausea—were more frequent in farmers those who had previously experienced an APP than in those who had not (Figure 1).

The frequency of symptoms was higher among farmers who had experienced an APP vs. those who had not (Figure 1, two-way ANOVA results *F* = 2.04, *p* < 0.001). The greatest absolute differences in symptom frequency between those who had experienced an APP and those who had not were observed for numbness or tingling in hands and feet (mean difference (MD): 0.93, 95% CI: 0.19, 1.66), insomnia (MD: 0.63, 95% CI: −0.10, 1.37), dizziness (MD: 0.61, 95% CI: −0.12, 1.35), more irritable or angry than normal (MD: 0.61, 95% CI: −0.12, 1.35), tired or low energy (MD: 0.54, 95% CI: −0.19, 1.28), shaking hands (MD: 0.52, 95% CI: −0.21, 1.25), Table A1). The only symptom that remained significant after adjustment for multiple testing was numbness or tingling in hands and feet.

After adjusting for age, having experienced at least one APP was associated with increased odds of fainting (OR: 7.48, 95% CI: 1.83, 30.74), shaking hands (OR: 3.50, 95% CI: 1.60, 7.60), numbness or tingling in hands or feet (OR: 3.23, 95% CI: 1.66, 6.32), insomnia (OR: 2.53, 95% CI: 1.34, 4.79), accelerated heart rate (OR: 2.42, 95% CI: 1.03, 5.47), dizziness (OR: 2.38, 95% CI: 1.19, 4.72), being more irritable or angry than normal (OR: 2.37, 95% CI: 1.23, 4.55), tired or low energy (OR: 2.33, 95% CI: 1.23, 4.46), and difficulty concentrating (OR: 2.01, 95% CI: 1.05, 3.85; Table 2).

In our sensitivity analysis, we found that farmers who applied/handled synthetic pesticides (e.g., conventional farmers) at the time of the interview reported a similar frequency of symptoms compared to non-applicators (e.g., organic farmers) (Figure 1, two-way ANOVA results *F* = 1.74, *p* = 0.19), suggesting that acute exposure to pesticides is unlikely to account for the difference in symptoms between those who had experienced an APP and those who had not. Only 4% (*n* = 12) of farmers had reported changing their pesticide use practices in the 12 months prior to interview.

### 3.4. Psychological Distress Symptoms During the Month Prior to Interview

Overall, BSI dimension-specific scores were similar in farmers who had experienced an APP compared to those who had not (Table 3). However, the farmers who reported experiencing an APP during the 10 years prior to interview (*n* = 10) had higher obsession-compulsion, depression, anxiety, hostility, phobic anxiety, paranoid ideation, and psychoticism BSI scores compared to farmers who had never suffered an event (Table 3). The latter findings are similar to those of a previous Costa Rican study of farmers who had experienced an APP in the one to three years prior to the interview (Table 3) [22]. Notably, our study population showed elevated scores in every BSI dimension compared with the BSI normative sample of 600 healthy German adults [33] (Appendix A).

No differences were observed in the odds of clinically relevant psychological distress in those who had previously experienced an APP after adjusting for age, education, and whether or not they applied/handled synthetic pesticides at the times of the interview (Table 4). However, farmers who reported having experienced an APP in the 10 years prior to interview experienced increased odds of abnormal BSI scores for hostility (OR: 4.51, 95% CI: 1.16, 17.70) and paranoid ideation (OR: 3.76, 95% CI: 0.99, 18.18).

## 4. Discussion

Our findings indicate that the prevalence of APP among farmers in Costa Rica is substantial (14%) and having experienced an APP may be associated with long-term neurological effects, particularly dizziness, feeling tired or low energy, experiencing an accelerated heart rate, difficulty concentrating, numbness or tingling in hands or feet, fainting, being more irritable or angry than normal, shaking hands, and insomnia. These results were similar to those of previous studies investigating associations between APPs and neurological symptoms [15,16,26,27]. These neurological symptoms are unlikely to be related to pesticide exposure at the time of the interview, as the sensitivity analysis showed that farmers who apply/handled pesticides at the time of the interview reported a similar frequency of symptoms to those who did not. The fact that only age remained predictive for having had an APP in the past, after adjustment for all other demographic and occupational factors, suggests that the dominant factor driving APPs in this population may be the total amount of time exposed to pesticides (age was highly correlated with years of pesticide use). Psychological distress symptoms in the month prior to the interview were not associated with ever having experienced an APP, but the odds of abnormal BSI scores for those who had experienced an APP in the 10 years prior to interview were elevated.

Consistent persistent symptom patterns among farmers who experienced at least one APP suggest that there are lasting health consequences of experiencing an event. The fact that this pattern occurs even among current non-applicators further implies that the symptoms are not related to recent pesticide application, but instead lasting consequences of APPs that may have occurred years before.

While having experienced an APP overall did not appear to be associated with psychological distress symptoms in the last month, that those who had experienced a pesticide poisoning in the past 10 years had elevated BSI scores in eight dimensions compared to the rest of the study population is suggestive that more recent APPs may be associated with psychological distress. However, this number of recent APPs (*n* = 10) is small, which prevents us from making strong conclusions about psychological distress in those with a recent APP. More research is warranted, but these results suggest that more recent pesticide poisonings may indeed affect mental functioning, and that this effect diminishes over time. This link between a recent APP and psychological distress as measured by the BSI was also shown in a cross-sectional 2010 study of banana workers in Costa Rica [22] and depressive symptoms were associated with a recent APP in farmers in other studies [20,21,38], providing further evidence for this hypothesis. It may also be that the association depends on the severity of the APP [38], which could not be measured by this study. Future studies should focus on psychological distress symptoms in those with recent APPs in LMICs, and especially addressing causality. In addition, our entire study population reported elevated psychological distress scores compared to the BSI normative population [33], suggesting that other factors may be impacting the neurological health of our study aside from pesticide use.

Another potential explanation for our findings may be that farmers that experienced an APP also experienced a higher rate of cumulative long-term exposure to pesticides over time, which may in turn lead to chronic health effects. Chronic long-term exposure to pesticides may be one explanation for the elevated reporting of psychological distress symptoms among the entire study population. Previous studies have reported that chronic high exposure to pesticides without an APP is also associated with depressive symptoms [15,20,28], but these studies did not also investigate organic farmers that had stopped applying conventional pesticides as in this case. More focused studies (e.g., case-control, cohort) are needed to establish causality. In addition, a better measurement of long-term exposure to pesticides is needed to assess how long-term exposure to pesticides may interact with APPs to cause chronic health effects. Traditional urinary biomarkers of pesticide exposure are not useful in the context of measuring long-term effects of APPs, as they have a short half-life of only a few days [39]. To better assess chronic health effects, questionnaire-based algorithms could be used to estimate cumulative exposure to multiple pesticides over time [40]. 

Having experienced an APP did not appear to affect pesticide use practice: those who had experienced an APP continued to use highly and moderately toxic pesticides similarly to those who had not experienced a pesticide poisoning. The fact that those who had experienced an APP were more likely to have received training on pesticide application is counterintuitive; it may be that farmers were more likely to seek out training on pesticide application after they had experienced an APP, or that since they are older on average they had more opportunities to have received training in pesticide application, or that pesticide use practices have become safer over time, especially as there is still considerable exposure to pesticides in the study population [4]. Other studies in LMICs suggest that risk of APP is independent of previous knowledge about pesticide application [12]. It may also reflect the changing demographics of APPs over time, with more Nicaraguan migrants experiencing APPs in the last 10 years.

Our study is limited by the self-reported nature of the APPs: only approximately a third were doctor-confirmed. However, previous studies have reported high reliability of self-reported APPs [16]. In addition, the proportion of farmers reporting an APP is very similar to other studies among agricultural workers in LMICs [6,14,15]. Further, previous research has found that minor cases of APP tend to be under-reported in self-reports [6]; this suggests that the true number of APP in this cohort is likely to be higher. This non-differential exposure misclassification in our cohort would tend to under-estimate the true effect of an APP on health outcomes. In addition, it is difficult to completely adjust our analyses for the effects of current and chronic exposure to pesticides and isolate the effect of APPs retrospectively given the cross-sectional nature of the data. While comparing the symptoms of current pesticide applicators with non-applicators helps to isolate the effects of APPs, residual confounding cannot be ruled out. The small numbers in many health outcome categories prevent us from drawing strong conclusions on the link between individual symptoms and APPs. Nevertheless, our results suggest a link between APP and chronic health effects. Future research should involve a prospective design that measures continued pesticide exposure, establishing a concrete timeline. Finally, tools such as the BSI should be normed for populations in low- and middle-income countries, as differences in clinical relevance with populations in Western Europe are likely.

## 5. Conclusions

Our study suggests that the burden of persistent adverse health effects related to APPs is substantial in a population of smallholder farmers in Costa Rica. Having experienced an acute pesticide poisoning may be associated with long-term health symptoms such as dizziness, feeling tired or low energy, experiencing an accelerated heart rate, difficulty concentrating, numbness or tingling in hands or feet, fainting, being more irritable or angry than normal, shaking hands, and insomnia. Additionally, more recent APPs may be linked to psychological distress symptoms. Research and interventions to prevent and treat APPs are needed, such as reducing use of pesticides by changing laws around pesticide use and distribution, increasing the use of protective equipment, following cohorts of APP-affected farmers over time, providing immediate medical assistance to affected farmers, and strengthening education programs.

## Figures and Tables

**Figure 1 ijerph-18-09021-f001:**
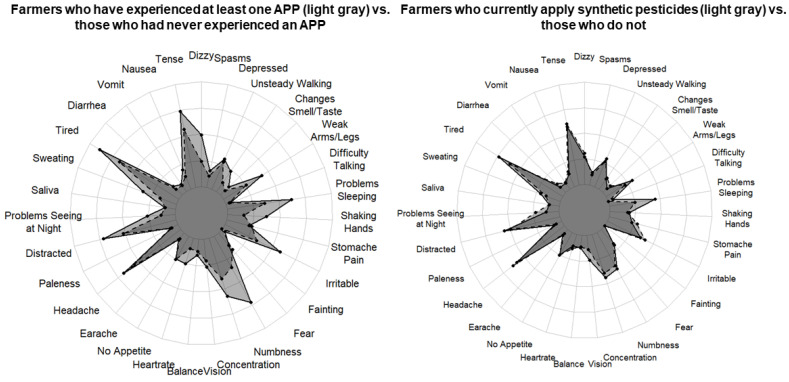
A comparison of the mean frequency of symptoms reported during the 12 months prior to interview by farmers who had experienced at least one APP (*n* = 43, in light gray) vs. those who had never suffered from an APP (*n* = 257, in dark gray) is on the left. As a sensitivity analysis, a comparison of the mean frequency of symptoms reported during the 12 months prior to interview by farmers currently applying synthetic pesticides (e.g., conventional farmers, *n* = 220, in light gray) vs. those who do not currently apply synthetic pesticides (e.g., organic farmers, *n* = 80, in dark gray) is on the right. Symptom frequency was rated on a scale from 1 (one time in the year) to 4 (more than once a week).

**Table 1 ijerph-18-09021-t001:** Socio-demographic and occupational characteristics of the study population (*n* (%) or median (P25–P75)), Zarcero County, Costa Rica, 2016, disaggregated by those who had experienced at least one acute pesticide poisoning (APP) and those who had not. World Health Organization (WHO) pesticide toxicity classifications were used.

Characteristics	Farmers Who Never Experienced APP (*n* = 257)	Farmers Who Experienced at Least One APP (*n* = 43)
Age (years)	31.0 (24.0–46.0)	50.0 (38.5–53.5)
Sex		
Male	231 (89.9)	42 (97.7)
Female	26 (10.1)	1 (2.3)
Marital status		
Married or living as married	150 (58.4)	33 (76.7)
Single	99 (38.5)	7 (16.3)
Separated/divorced/widowed	8 (3.1)	3 (7.0)
Job position		
Farm owner	86 (33.5)	27 (62.8)
Farmworker	171 (66.5)	16 (37.2)
Country of birth		
Costa Rica	145 (56.4)	32 (74.4)
Nicaragua	112 (43.6)	11 (25.6)
Education (years)	6.0 (5.0–8.0)	6.0 (6.0–6.0)
Smoking habits		
Never	129 (50.2)	21 (48.8)
Former	70 (27.3)	17 (39.5)
Current	58 (22.6)	5 (11.6)
Number of drinks per week	0.0 (0.0–4.0)	0 (0.0–0.0)
Time applying pesticides (years)	14.0 (7.0–28.0)	34.0 (23.5–38.5)
Applicator of synthetic pesticides at time of interview		
No	68 (26.5)	12 (27.9)
Yes	189 (73.5)	31 (72.1)
Household income (colones)	320,000 (260,000–500,000)	320,000 (228,000–600,000)
Household size	4.0 (3.0–5.0)	4.0 (2.5–5.0)
Training in safe practices for pesticide use	98 (45.4)	25 (65.8)
No	118 (45.9)	13 (30.2)
Yes	98 (45.4)	25 (65.8)
Pesticide use at time of interview		
Time spent using pesticides (hours/week)	8.0 (4.0–13.0)	8.0 (4.0–21.0)
Used extremely or highly toxic pesticides in the last week (WHO classification Ia or Ib)		
No	200 (77.8)	37 (86.0)
Yes	34 (14.5)	4 (9.8)

**Table 2 ijerph-18-09021-t002:** Frequency of neurological symptoms in the last 12 months associated with having experienced at least one APP (predictor) in male farmers (*n* = 273) in an ordinal logistic regression model. Symptom frequency was rated on a scale from 0 (never) to 4 (more than once a week). Odds ratios (OR) and their associated 95% confidence intervals (CI) are reported.

Symptom (Outcome)	Unadjusted OR (95% CI)	Adjusted OR (95% CI)
Fainting	4.99 (1.38, 17.40)	7.48 (1.83, 30.74)
Shaking hands	2.39 (1.17, 4.75)	3.50 (1.60, 7.60)
Numbness or tingling in hands or feet	3.30 (1.73, 6.27)	3.23 (1.66, 6.32)
Insomnia	2.44 (1.32, 4.47)	2.53 (1.34, 4.79)
Accelerated heart rate	2.38 (1.06, 5.09)	2.42 (1.03, 5.47)
Dizziness	2.59 (1.33, 5.01)	2.38 (1.19, 4.72)
More irritable or angry than normal	2.22 (1.19, 4.14)	2.37 (1.23, 4.55)
Tired or low energy	2.10 (1.14, 3.91)	2.33 (1.23, 4.46)
Excessive salivation	1.21 (0.39, 3.14)	2.27 (0.67, 6.79)
Changes in ability to smell and taste	1.86 (0.64, 4.71)	2.23 (0.73, 6.17)
Difficulty concentrating	1.91 (1.02, 3.53)	2.01 (1.05, 3.85)
Excessive sweating/perspiration	1.90 (0.85, 3.99)	1.89 (0.82, 4.11)
Headache	1.52 (0.83, 2.78)	1.88 (0.99, 3.55)
Distracted, forgetful, or confused	2.05 (1.11, 3.82)	1.87 (0.99, 3.54)
Vomiting	1.18 (0.42, 2.88)	1.74 (0.58, 4.61)
Difficulty seeing at night	1.97 (0.85, 4.26)	1.67 (0.70, 3.75)
Unsteady walking	1.93 (0.77, 4.43)	1.59 (0.62, 3.73)
Tense, anxious or nervous	1.90 (1.04, 3.48)	1.57 (0.84, 2.94)
Nausea	1.74 (0.81, 3.56)	1.55 (0.70, 3.28)
Difficulty talking	1.20 (0.27, 3.87)	1.49 (0.32, 5.31)
Paleness	1.11 (0.36, 2.85)	1.41 (0.44, 3.90)
Loss of hunger/appetite	1.10 (0.50, 2.25)	1.40 (0.62, 3.00)
Weakness in arms and legs	1.68 (0.84, 3.25)	1.38 (0.68, 2.74)
Earache	1.18 (0.42, 2.87)	1.33 (0.46, 3.42)
Diarrhea	1.23 (0.52, 2.66)	1.12 (0.46, 2.49)
Abdominal or stomach pain	0.97 (0.46, 1.94)	1.11 (0.51, 2.32)
Blurred or double vision	1.46 (0.70, 2.90)	1.02 (0.47, 2.09)
Difficulty maintaining equilibrium	1.22 (0.43, 2.97)	0.99 (0.34, 2.47)
Felt scared/fear	0.85 (0.38, 1.76)	0.88 (0.38, 1.87)
Depressed, indifferent, or withdrawn	0.99 (0.49, 1.93)	0.87 (0.42, 1.72)

Models were adjusted for age and restricted to men only, as only one woman experienced an APP.

**Table 3 ijerph-18-09021-t003:** Median (interquartile range (IQR)) Brief Symptom Inventory (BSI) descriptive parameters in farmers who reported a pesticide poisoning vs. those who did not. A higher score corresponds to a higher frequency of symptoms in that dimension in the past month in the study population. The last column compares the data from this study with that of a different 2010 study in Costa Rican farmers that administered the BSI to farmers that had experienced an acute pesticide poisoning (APP) in the previous one to three years [22].

BSI Dimensions	Never Experienced APP (*n* = 257)	Experienced at Least One APP during Their Lifetime (*n* = 43)	Experienced at Least One APP during the Last 10 Years (*n* = 10)	Farmers from a 2010 Costa Rican Study Who had Experienced one APP 1–3 Years Previously (*n* = 43) [22]
Somatization	0.29 (0.0, 0.57)	0.29 (0.07, 0.64)	0.36 (0.07, 1.11)	0.86 (0.08, 1.65)
Obsession-compulsion	0.50 (0.0, 1.0)	0.50 (0.17, 1.0)	0.75 (0.17, 1.46)	0.83 (0.17, 1.50)
Interpersonal sensitivity	0.25 (0.0, 0.75)	0.25 (0.0, 0.75)	0.25 (0.06, 0.94)	0.50 (0.0, 1.0)
Depression	0.17 (0.0, 0.67)	0.17 (0.0, 0.5)	0.50 (0.50, 0.67)	0.33 (0.0, 1.0)
Anxiety	0.17 (0.0, 0.67)	0.33 (0.0, 0.75)	0.58 (0.08, 1.17)	0.40 (0.0, 1.0)
Hostility	0.20 (0.0, 0.4)	0.20 (0.0, 0.4)	0.60 (0.20, 1.15)	0.40 (0.0, 0.80)
Phobic anxiety	0.0 (0.0, 0.4)	0.0 (0.0, 0.4)	0.40 (0.20, 0.90)	0.40 (0.0, 1.0)
Paranoid ideation	0.80 (0.2, 1.2)	0.80 (0.2, 1.2)	1.00 (0.85, 1.40)	0.0 (0.0, 0.8)
Psychoticism	0.20 (0.0, 0.8)	0.20 (0.0, 0.6)	0.60 (0.40, 1.25)	0.25 (0.0, 0.75)
Global Severity Index (GSI)	0.34 (0.11, 0.75)	0.38 (0.18, 0.62)	0.59 (0.42, 0.97)	0.71 (0.28, 1.15)

**Table 4 ijerph-18-09021-t004:** Prevalence and odds of cases of mental health according to normalized T-scores on the Brief Symptom Inventory (BSI) (cutoff T-score ≥63) in the study population.

BSI Dimensions	Never Experienced APP (*n* = 257)		Experienced at least One APP during Their Lifetime (*n* = 43)		Experienced at least One APP during the Last 10 Years (*n* = 10)	
	Prevalence *n* (%)	OR (95% CI)	Prevalence *n* (%)	OR (95% CI)	Prevalence *n* (%)	OR (95% CI)
Somatization	78 (30.4)	1.0	12 (27.9)	1.08 (0.49, 2.26)	4 (40.0)	1.57 (0.38, 5.88)
Obsession-compulsion	61 (23.7)	1.0	9 (20.9)	1.05 (0.44, 2.33)	3 (30.0)	1.38 (0.29, 5.22)
Interpersonal sensitivity	53 (20.6)	1.0	10 (23.3)	1.57 (0.67, 3.53)	3 (30.0)	1.80 (0.37, 7.03)
Depression	57 (22.2)	1.0	7 (16.3)	0.82 (0.31, 1.92)	2 (20.0)	0.91 (0.13, 3.94)
Anxiety	50 (19.5)	1.0	11 (25.6)	1.47 (0.65, 3.17)	3 (30.0)	2.0 (0.41, 7.65)
Hostility	50 (19.5)	1.0	7 (16.3)	1.25 (0.46, 3.06)	5 (50.0)	4.51 (1.16, 17.70)
Phobic anxiety	55 (21.4)	1.0	8 (18.6)	1.10 (0.44, 2.53)	4 (40.0)	2.55 (0.62, 9.52)
Paranoid ideation	101 (39.3)	1.0	19 (44.2)	1.55 (0.77, 3.11)	7 (70.0)	3.76 (0.99, 18.18) *
Psychoticism	77 (30.0)	1.0	9 (20.9)	0.87 (0.36, 1.92)	5 (50.0)	2.48 (0.65, 9.45)
Global Severity Index (GSI)	81 (31.5)	1.0	11 (25.6)	0.91 (0.41, 1.92)	5 (50.0)	2.16 (0.57, 8.24)

* Borderline statistically significant.

## Data Availability

The data presented in this study are available on request from the corresponding author with permission from the participating institutions. The data are not publicly available due to data ownership and privacy issues.

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
