# Peer review of "Long-Term Neurological and Psychological Distress Symptoms among Smallholder Farmers in Costa Rica with a History of Acute Pesticide Poisoning"

_ijerph, 2021, doi:10.3390/ijerph18179021_

Round 1

Reviewer 1 Report

The paper is very interesting, it is well writing and demonstrates important findings to the field involving pesticides exposure and chronic effects to the health of farmers. I inserted some comments and requests in the paper file which is uploaded. Please, see the paper below.

Author Response

Reviewer 1

The paper is very interesting, it is well writing and demonstrates important findings to the field involving pesticides exposure and chronic effects to the health of farmers. I inserted some comments and requests in the paper file which is uploaded. Please, see the paper below.

Response: Thank you for your positive words about our paper, and for your thorough reading of our analysis.

Lines 112-113: Excellent and important information! Such information is extremely important for this kind of paper. Congratulations!

Response: We agree that these variables are particularly important for this study. Thank you again for your positive evaluation.

Line 130: Can the authors explain such statement, please?

Response: Thank you for this question. We based our analysis on self-reported APPs as opposed to symptoms because it is very difficult to accurately classify APPs after the fact, as pointed out by the WHO classification tool for pesticide poisonings. By relying on self-reported APPs, we were able to identify APPs that had occurred far in the past, and explore the long-term symptoms associated with these APPs. To clarify this point further, we have revised the sentence to read as follows: “Symptom occurrence was not used to define whether they had an APP, as clinical presentation can vary widely based on the poisoning agent and the non-specificity of symptoms makes it difficult to definitively diagnose all cases of APP [2].”

Line 179: Can the authors explain the reason to use such cut-off, please.

Response: We used this cut-off because it is listed as the clinically relevant cut-off in the BSI manual. We revised this sentence to make this more clear, “To model whether having experienced an APP predicted clinically significant psychological distress based on BSI scores, we fitted logistic regression models for each symptom dimension and the GSI using dichotomized T-scores representing clinically relevant distress (1: T-score≥63; 0: T-score<63).”

Line 192: Very interesting and important result!

Response: Thank you, we also were struck by the fact that all the sufferers of APPs had been working on conventional farms.

Line 370: And also more studies that could follow such population in an interesting period as 10 years of activity or more, if possible.

Response: Thank you for this important point. We have revised this sentence to include it as follows: “Research and interventions to prevent APPs by reducing use of pesticides, increasing the use of protective equipment, following cohorts of APP-affected farmers over time, and strengthening education programs are needed.”

Reviewer 2 Report

The manuscript is of high interest in relation to the damage produced by exposure to pesticides. In particular, the novelty lies in having included organic farmers, however they were not separated in the analysis.

Although the comparison with a previous study is interesting, I consider that these results should not be presented in Table 3, but only discussed in the discussion section. In this way, Table 3 could be simplified and show the results of scores, instead of the interquartile range, which is easier to interpret. 

About changes in the way of use of pesticides (lines 254-255), how is it related to organic producers?

In the description of table 4 (line 277) include a comment that paranoid ideation is not statistically significant or is on the edge of significance.

The legend of table 1 of the appendix incorporates information from Figure 2. I suggest to correct so that the proper explanation remains in the figure (about the gray tones of the graph). In addition, this would be figure 1 of the appendix, although the second in the article. 

I assume that it is the only reference available, I wonder if taking the German population as a reference is adequate when compared to the Latino population.

Author Response

Reviewer 2

 The manuscript is of high interest in relation to the damage produced by exposure to pesticides. In particular, the novelty lies in having included organic farmers, however they were not separated in the analysis.

Response: Thank you for your consideration and positive evaluation of our study. While it is true that we were interested in the long-term symptoms associated with APPs in both conventional and organic farmers, we were also interested in the differences in symptoms between those currently applying synthetic pesticides (usually conventional farmers) and those not applying synthetic pesticides (usually organic farmers) and explored those differences in Figure 1. As you can see from Figure 1, there is a difference in frequency of symptoms among those who have or have not had an APP, but virtually no difference in symptoms between those currently applying synthetic pesticides and those not. To make this more clear, we have revised the legend for this figure to mention the terms “conventional” and “organic” farming.

Although the comparison with a previous study is interesting, I consider that these results should not be presented in Table 3, but only discussed in the discussion section. In this way, Table 3 could be simplified and show the results of scores, instead of the interquartile range, which is easier to interpret.

Response: Thank you for this point. It is certainly an option to move it only to the discussion section, but we are unsure what is meant by reporting the results of scores. Here we reported the median and interquartile range; we could have also presented the mean and standard deviation, but it does not change the interpretation substantially. Therefore we propose leaving it to the editor as to whether we should eliminate this comparison with a previous study. We could also present the numbers of clinically relevant cases, but this provides less information than looking at the scores directly in terms of increased incidence of distress.

About changes in the way of use of pesticides (lines 254-255), how is it related to organic producers?

Response: This is a very good question, and has been partially addressed above. The reason we discuss “applicators” vs “non-applicators” here is for increased specificity over the terms “conventional” vs “organic,” as there were in fact some conventional workers that no longer applied pesticides. But these terms are largely overlapping, and we have reintroduced them here for clarity and continuity with the introduction: “In our sensitivity analysis, we found that farmers who applied/handled synthetic pesticides (e.g. conventional farmers) at the time of the interview reported a similar frequency of symptoms compared to non-applicators (e.g. organic farmers) (Figure 1, two-way ANOVA results F=1.74, p=0.19), suggesting that acute exposure to pesticides is unlikely to account for the difference in symptoms between those who had experienced an APP and those who had not.”

In the description of table 4 (line 277) include a comment that paranoid ideation is not statistically significant or is on the edge of significance.

Response: Thank you: we have done so.

The legend of table 1 of the appendix incorporates information from Figure 2. I suggest to correct so that the proper explanation remains in the figure (about the gray tones of the graph). In addition, this would be figure 1 of the appendix, although the second in the article.

Response: Thank you; we have corrected these two legends.

I assume that it is the only reference available, I wonder if taking the German population as a reference is adequate when compared to the Latino population.

Response: We agree that this reference population is not ideal, but unfortunately it is the only one available to us. This is one reason that we included the previous study in Costa Rica that used the BSI as a comparison in Table 3. We added a line (363-365) to the limitations to this effect: “Finally, tools such as the BSI should be normed for populations in low- and mid-dle-income countries, as differences in clinical relevance with populations in Western Europe are likely.”

Reviewer 3 Report

The manuscript entitled „”Long-term neurological and psychological distress symptoms among smallholder farmers in Costa Rica with a history of acute pesticide poisoning” is generally well written with mostly actual references cited (1993-2020). The acute poisoning of pesticides is still utmost topic of the very high concern not only reflecting to the farmers in Costa Rica but every worker exposed occupationally.

Special comments:

  1. In the Introduction section the Authors focused on poisoning in low- and middle-income countries. I agree that this is huge problem so it should be highlighted. I would suggest to re-write Introduction section, adding that farmers are occupationally exposed to pesticides, including regulations among workers and pesticides with comparing it with European Union Law.
  2. It would be better if Materials and Methods section were described in detail without sending to other articles. In addition, in line 83 („300 smallholder farmers”) should be added information concerning sex and age of participants (300 smallholder farmers (273 men, 27 women) aged i.e. 24 – 53…)
  3. In the article, the Authors write about two types of farm: conventional and organic. I am not sure what is the difference between them? Does the second one not use pesticides?
  4. Does the questionnaire have a question concerning drinking alcohol? What about co-existing chronic diseases which may have have affect on psychological distress and other health effects. In addition, in this section should be also added sentence „The details can be found in Table 1.”
  5. Below Table 2 The Authors add the footnote that „models were adjusted for age and restricted to MEN only”. My question is why? Maybe it is related to majority of the of the studied population? This difference should be clarified. Please do not forget about data in Results and Abstract sections.
  6. The data from Table 3 and 4 are also concerned only to men farmers?
  7. In Discussion section, I recommend to divide acute from chronic exposure.
  8. In Conclusions, I suppose that additional information will be needed: change of law, immediate medical help especially in case of acute pesticide poisoning, biological monitoring preventing against long-term pesticide critical effects to human health, etc.

Author Response

Reviewer 3:

The manuscript entitled „”Long-term neurological and psychological distress symptoms among smallholder farmers in Costa Rica with a history of acute pesticide poisoning” is generally well written with mostly actual references cited (1993-2020). The acute poisoning of pesticides is still utmost topic of the very high concern not only reflecting to the farmers in Costa Rica but every worker exposed occupationally.

Response: Thank you for these comments and reflections, we agree that this topic is of high importance globally.

Special comments:

    In the Introduction section the Authors focused on poisoning in low- and middle-income countries. I agree that this is huge problem so it should be highlighted. I would suggest to re-write Introduction section, adding that farmers are occupationally exposed to pesticides, including regulations among workers and pesticides with comparing it with European Union Law.

Response: Thank you for this suggestion. This is an important point, and we have added that farmers are occupationally exposed to pesticides in line 41 of the introduction (“Occupational exposure to pesticides is ubiquitous among farmers worldwide [1] and its adverse effects on human health are a well-documented threat [2, 3].”). The inclusion of regulations among workers varies widely between countries in LMICs, and it would be a paper in and of itself to describe these variations (see the first paper citation for an example). To attempt to sum up this landscape briefly, we added the following sentence: “While European countries have developed legislation to restrict the use of hazardous pesticides such as glyphosate, the use of pesticides in developing countries has grown rapidly without similar regulation [1].” (lines 50-52) 

    It would be better if Materials and Methods section were described in detail without sending to other articles. In addition, in line 83 („300 smallholder farmers”) should be added information concerning sex and age of participants (300 smallholder farmers (273 men, 27 women) aged i.e. 24 – 53…)

Response: Thank you for this point. We have added details about the sex and age of the participants to the methods, although normally we would add this information only to the results. We leave this up to the discretion of the editor whether it should stay.

We absolutely agree that the materials and methods need to be described in detail without referring to other articles; however, in this case, the PESTROP study represented a large study of which this analysis is just one small facet. Therefore we describe here only the methods relevant to this specific article, and refer to the published study protocol for information about the wider study. It would enormously increase the length of the article to describe instead the full methods of all aspects of the study.

    In the article, the Authors write about two types of farm: conventional and organic. I am not sure what is the difference between them? Does the second one not use pesticides?

Response: Thank you, that is exactly right. We clarify this in lines 70-73 of the methods: “By comparing farmers who switched from conventional farming practices (i.e., intensive use of synthetic pesticides) to organic farming practices (i.e. no use of synthetic pesticides), it is possible to assess whether lingering symptoms of past APPs remain even in those who no longer use synthetic pesticides.”

    Does the questionnaire have a question concerning drinking alcohol? What about co-existing chronic diseases which may have have affect on psychological distress and other health effects. In addition, in this section should be also added sentence „The details can be found in Table 1.”

Response: Yes, the questionnaire does have a question regarding drinking alcohol. We have added a line to table 1 including the results and the differences between those who did and did not suffer an APP. The reason that we did not include this variable initially is that most participants did not report drinking very much on average, or did not answer the question on frequency (see Table 1).

Participants were asked about respiratory conditions such as asthma, eczema, diabetes, hypertension, high cholesterol, head injuries, and “other” illnesses, but it is difficult to say whether some of these conditions might be on the causal pathway between the previous APP and lingering symptoms. In addition, up to 50% of participants reported not knowing if they had one of these conditions, making the data very difficult to interpret. Therefore we decided to omit these data from this paper.

We have added the line you suggested to the methods (lines 117-118).

    Below Table 2 The Authors add the footnote that „models were adjusted for age and restricted to MEN only”. My question is why? Maybe it is related to majority of the of the studied population? This difference should be clarified. Please do not forget about data in Results and Abstract sections.

Response: Thank you for this excellent question. Yes, we would have preferred to adjust by sex, but as only one woman experienced an APP the numbers were just too small to be able to be able to make statements specific to women who experienced APPs. We have added clarifying points highlighting this to the abstract (line 28), the results (line 198), and to the table footnote (lines 254-255).

    The data from Table 3 and 4 are also concerned only to men farmers?

Response: No, these tables include data from all farmers that report an APP, including the one woman.

    In Discussion section, I recommend to divide acute from chronic exposure.

Response: Thank you for this suggestion. We have separated out the part discussing the potential for chronic exposure into its own paragraph (lines 340-352).

    In Conclusions, I suppose that additional information will be needed: change of law, immediate medical help especially in case of acute pesticide poisoning, biological monitoring preventing against long-term pesticide critical effects to human health, etc.

Response: Thank you for these excellent additional points. We have added them to lines 390-395 of the conclusions.

Round 2

Reviewer 3 Report

The Authors have re-written the manuscript in accordance to the Reviewer’s suggestions which improved its quality. Tha article has better methodological background and discussion sections. Missing data in Materials and Mathods sections has been added which clarify received data. The Authors’ Conclusions are appropriate which makes it valuable contribution to the journal and interesting to the readers.